# Enhancing Encapsulation Efficiency of Chavir Essential Oil via Enzymatic Hydrolysis and Ultrasonication of Whey Protein Concentrate–Maltodextrin

**DOI:** 10.3390/foods13091407

**Published:** 2024-05-03

**Authors:** Nasrin Beigmohammadi, Seyed Hadi Peighambardoust, Asad Mohammad Amini, Kazem Alirezalu

**Affiliations:** 1Department of Food Science, College of Agriculture, University of Tabriz, Tabriz 5166614766, Iran; nasrinbeigmohammadi@yahoo.com (N.B.); kazem.alirezalu@tabrizu.ac.ir (K.A.); 2Department of Food Science and Engineering, Faculty of Agriculture, University of Kurdistan, Sanandaj 6617715175, Iran; a.mohammadamini@uok.ac.ir

**Keywords:** core material, chavir essential oil, encapsulation, hydrolysis, ultrasonication, stability

## Abstract

This study focused on the characterization of emulsions and microparticles encapsulating Chavir essential oil (EO) by application of modified whey protein concentrate–maltodextrin (WPC-MD). Different physical, chemical, morphological, thermal, and antioxidant properties and release behavior of spray-dried microparticles were assessed. Antioxidant, solubility, emulsifying, and foaming activities of modified WPC were increased compared to those of primary material. The results indicated that the particle size distribution varied depending on the type of carriers used, with the smallest particles formed by hydrolyzed WPC (HWPC). Binary blends of modified WPC-MD led to improved particle sizes. The spray-drying yield ranged from 64.1% to 85.0%, with higher yields observed for blends of MD with sonicated WPC (UWPC). Microparticles prepared from primary WPC showed irregular and wrinkled surfaces with indentations and pores, indicating a less uniform morphology. The UWPC as a wall material led to microparticles with increased small cracks and holes on their surface. However, HWPC negatively affected the integrity of the microparticles, resulting in broken particles with irregular shapes and surface cracks, indicating poor microcapsule formation. Encapsulating EO using WPC-MD increased the thermal stability of EO significantly, enhancing the degradation temperature of EO by 2 to 2.5-fold. The application of primary WPC (alone or in combination with MD) as wall materials produced particles with the lowest antioxidant properties because the EO cannot migrate to the surface of the particles. Enzymatic hydrolysis of WPC negatively impacted microparticle integrity, potentially increasing EO release. These findings underscore the crucial role of wall materials in shaping the physical, morphological, thermal, antioxidant, and release properties of spray-dried microparticles, offering valuable insights for microencapsulation techniques.

## 1. Introduction

*Ferulago angulata* is a perennial plant with small yellow flowers locally known as ‘Chavir’ or ‘Chavil’, which grows in the western provinces of Iran (Kermanshah and Kurdistan) [1]. Traditionally, the aerial parts of the plant are added to edible fats, butter, and meat products by locals to increase their shelf life while also enhancing their flavor [2]. The essential oil (EO) extracted from the flowers of Chavir is a rich source of bioactive compounds with potent antioxidant and antibacterial properties, in addition to other medicinal benefits [3,4]. Based on reports in the literature, these components mostly belong to monoterpenes and sesquiterpenes categories. More specifically, compounds such as α-pinene, α-phellandrene, β-ocimene, p-cymene, methyl carvacrol, myrcene, and terpinolene have been reported in the EO from *Ferulago angulata* [5]. In general, EOs are highly concentrated extracts that contain volatile compounds responsible for the characteristic fragrance and therapeutic properties of the plant. Some biological activities such as antimicrobial, antioxidant, and therapeutic properties such as anthelmintic, antifungal, and antiviral have been reported for EO from Chavir [3,6,7]. The concentration of EO in aromatic herbal plants can vary depending on factors such as the plant species, growing conditions, and extraction method. According to previous reports, the EO yield of the aerial parts and seeds of *Ferulago angulata* collected from western Iran (Kermanshah province) was 0.63 and 3.2% (*v*/*w*) based on the dry weight, respectively [3].

Despite the beneficial biological properties of essential oils, their direct application in food can pose challenges. High concentrations of EOs used to leverage their antimicrobial effects can lead to undesirable organoleptic effects in food [8]. To address this issue, researchers have explored the encapsulation of EOs in stable micro- or nano-particles as a viable approach to enhance their utilization, stability, and efficacy [9]. Moreover, due to the volatile nature and the instability of EO components, it is necessary to preserve them against possible adverse environmental or processing conditions while maintaining their bioavailability via microencapsulation. There are different microencapsulation techniques proposed with their benefits and influential parameters [10]. Spray-drying is the prevalent method used for solidifying emulsions using suitable carriers based on proteins and polysaccharides in their intact or modified forms [11]. A decade overview and prospect of spray drying encapsulation of bioactive compounds from fruit products showed that the type and the ratio of encapsulating material had a significant impact on the physiochemical properties of the microcapsules, and spray drying encapsulation could open up new possibilities for the controlled delivery of beneficial compounds to the host [12].

Whey proteins are suitable material for the microencapsulation of bioactive compounds via a spray-drying process. However, low solubility, aggregation, agglomeration, sedimentation, and denaturation of these proteins under acidic conditions and/or under mechanical and thermal stress may cause loss of their functional properties and limit their use as an emulsifier and carrier [13]. Therefore, modification of proteins’ structure by non-thermal techniques, such as high-pressure processing (HPP), pulsed electric field (PEF), cold plasma, irradiation, and ultrasonication, have been proposed [14,15]. The application of high-power ultrasound would enhance the surface activity of most proteins, leading to improved foaming and emulsifying properties [16]. Partial hydrolysis of proteins produced hydrolysates or peptides with enhanced functional characteristics and biological activities [17]. Proteases break peptide bonds and liberate low molecular weight protein fractions with improved solubility and biological and surface activities, as compared to their primary proteins [18]. This structural modification and improved functional properties of protein hydrolysates positively affect their amphiphilic characteristics by their tendency to interact at the water/oil interfaces, leading to stabilizing emulsions [19]. Moreover, as a result of enzymatic, structural changes in proteins, they effectively show film-forming properties that can be utilized as wall materials for the microencapsulation of many foods’ bioactive components. Maltodextrin (MD), a commonly used food additive, is among the wall materials that are used for spray-drying microencapsulation of active bioactive substances due to its availability and lower price, biocompatibility, biodegradability, and non-toxicity. The combination of MD with other hydrocolloids, gums, and protein hydrolysates can positively affect the physicochemical characteristics of obtained microparticles. In this respect, spray-drying encapsulation efficiency and release properties of particles can be improved [20].

Nevertheless, there is limited research on the application of enzymatically or power ultrasound-modified WPC as wall materials for spray-drying EO-loaded microparticles. One of the primary goals of this study was to use binary blends of MD with WPC that are modified by enzymatic hydrolysis and ultrasonication as carriers in spray-drying microencapsulation. The combination of MD with protein hydrolysates can positively affect the physicochemical characteristics of obtained microparticles. Therefore, the main objectives of this study were the following: (1) to evaluate the effectiveness of physically or enzymatically modified WPC and its combination with MD as carriers for microencapsulation of herbal essential oil during spray-drying; (2) to examine the properties of emulsions produced from the binary blends of modified WPC with MD; and (3) to explore the diverse technological and functional attributes of spray-dried powders containing the essential oil.

## 2. Materials and Methods

### 2.1. Materials

Aerial parts (leaf and stem) of Chavir (*Ferulago angulate*) were collected from the Dallahou mountains (Kermanshah, Iran) by three local farmers over two days. The plant’s authenticity was confirmed by expert botanists from the University of Kurdistan (Sanandaj, Iran). The freshly collected Chavir was then dried in the shade in preparation for essential oil extraction. MD with DE of 15–20 (18.8 DE, test result by provider) and WPC (Hilmar 9010, 92.0% protein, dry basis) were purchased from Dalian Future International Co. (Dalian, China) and Hilmar Ingredients (Hilmar, CA, USA), respectively. The Alcalase enzyme from *Bacillus licheniformis* (specific activity ≥ 0.75 Anson units/mL) was acquired from Sigma-Aldrich (St. Louis, MI, USA), and other chemicals were provided from Merck (Darmstadt, Germany).

### 2.2. Extraction and Characterization of EO

A Clevenger-type apparatus applying the steam-distillation method was used to extract the EO from the aerial parts of the plant according to the detailed method reported previously [21]. The collected EO (1.2 ± 0.1% *v*/*w*, wet basis) was purified following freezing at −20 °C and stored in airtight dark containers for subsequent analysis. A GC-MS system (Agilent 7890A gas chromatography with 5975C mass spectrometer, Agilent Technologies, Santa Clara, CA, USA) equipped with a DB-35MS column (30 m long, ID 0.32 mm) was employed to identify the chemical constituents of the EO according to the method reported previously [6].

### 2.3. Whey Protein Modification

The enzymatic hydrolysis of 10% (*w*/*v*) whey protein aqueous solution in potassium phosphate buffer (pH = 7.6) was performed using Alcalase enzyme (100:2 ratio) at a constant temperature of 38 °C for 60 min under continuous stirring (300 rpm). Following hydrolysis, the enzyme activity was halted by placing the reaction chamber in a hot water bath at 96 °C for 20 min. This sample is encoded as HWPC (hydrolyzed WPC). The obtained mixture was centrifugated at 8000× *g* for 15 min, and the supernatant comprising hydrolysates was collected and subjected to freeze-drying until subsequent use. For power ultrasonic treatments, 100 mL of WPC (5% *w*/*v*) in phosphate buffer (pH adjusted to 7.4) was sonicated using a probe-type device (UHP-400, Hielscher, Teltow, Germany) with a titanium probe (diameter 13 mm) with operating power of 400 W and frequency of 20 kHz with a pulsed mode of 1.0 s on/1.0 s off for 30 min. The solution temperature was kept below 15 °C using an ice bath during the sonication. The sonicated WPC solution (called UWPC) was freeze-dried and stored frozen until the next use.

### 2.4. Emulsion Preparation and Characterization

Different formulations of wall materials and their binary systems were prepared according to Table 1.

The total solids content of emulsions was fixed at 30% (for use as feed for spray-drying), in which the binary systems were mixed at a 1:1 ratio. The emulsion preparation procedure consisted of adding wall material including MD, WPC, hydrolyzed WPC (H-WPC), ultrasound-treated WPC (U-WPC), or their blends to distilled water. The solution was brought to continuous stirring at ambient temperature for 30 min, shaking for 24 h using an orbital shaker at room temperature. Then, an appropriate amount of EO (based on EO to WM ratios suggested in Table 1) was added to the latter solution, and it was homogenized at 7500 rpm for 4 min using Ultra-Turrax T18 digital (IKA, Staufen Im Breisgau, Germany). The sample codes for emulsions are mentioned in Table 1. The stability of emulsions, as a critical measure for spray-drying, was measured by the creaming index (CI) according to the detailed description by Frascareli et al. [22]. The droplet size and Zeta potential of emulsions were characterized based on the dynamic light scattering (DLS) method using Zetasizer Nano ZS (Malvern Panalytical, Malvern, UK) at 24 °C. The average droplet size was reported as the Z-average diameter of particles.

### 2.5. Spray-Drying Microencapsulation

A pilot-scale spray dryer (Maham Sanaat Co., Neyshabur, Iran) was used to solidify prepared emulsions. The spray-drier chamber (100 cm diameter × 200 cm height) was used with a rotary atomizer (5 cm diameter) in a co-current flow layout. A volumetric feed pump (fluid capacity of 3 L/h and a pressure of 1 bar) was used to feed the atomizer. In preliminary experiments, drying parameters were examined, and the best drying parameters were selected as follows. The inlet and outlet air temperatures were set to 160 ± 5 °C and 80 ± 5 °C, respectively. Atomizer rotational speed of 15,000 rpm, feed flow rate of 10 mL/min, feed temperature of 30 ± 2 °C, and atomizer pressure of 4.5 ± 0.2 bar were kept constant during all spray-drying trials. Based on the preliminary results, emulsions prepared with higher ratios of essential oil/wall materials (EO:WM) showed larger droplets with all wall materials used compared to their corresponding samples with lower EO concentrations. For this reason, spray-drying treatments were carried only with feed emulsions of lower EO content (EO:WM ratio of 1:3). This feed composition was kept constant for all spray-drying treatments. Dried emulsion powders were collected from the bottom of the drying chamber and kept in air-tight containers inside the refrigerator (4 ± 0.1 °C) until used. The weight ratio of spray-dried powders to the initial solid contents in the feed solution of the emulsions was stated as production yield.

### 2.6. Droplet and Particle Size and Zeta Potential

The droplet/particle sizes and the Zeta potentials of emulsions and spray-dried microparticles were assessed using the dynamic light scattering (DLS) technique with a Zetasizer (Nanosizer 3000, Malvern Instruments, Malvern, UK) at ambient temperature with an angle of 90°. Prior to measurements, the samples were diluted 100-fold with distilled water. The experiments were conducted in triplicate, and the volume mean diameter was utilized to indicate the average emulsion/particle size.

### 2.7. Encapsulation Efficiency of Emulsions and Spray-Dried Microparticles

The essential oil impregnated inside emulsion droplets or spray-dried microspheres is regarded as encapsulation efficiency (EE). To measure EE of emulsions, aliquots of 5 mL emulsion and n-hexane were mixed under magnetic stirring for 30 min to remove the EO content on the surface of droplets. The mixture was then centrifugated for 3 min at 3000 rpm, and the solvent phase was discarded. This procedure was repeated twice. In the next step, the obtained emulsion was mixed with 10 mL deionized water and mixed for 5 min, blended with 5 mL n-hexane while stirring for 5 min, followed by centrifugating at 5000 rpm for 10 min. The solvent phase containing EO was collected, and its absorbance was determined at 270 nm using a UV-2100 spectrophotometer (UNICO, Franksville, WI, USA). The concentration of EO was calculated based on a calibration curve, and the percentage ratio of the loaded EO to its initial concentration was determined as the encapsulation efficiency (EE) of the emulsions [23]. For the spray-dried particles, the same procedure was followed, except that 500 mg of powder was used instead of 5 mL of emulsion [24].

### 2.8. Density, Flowability, and Cohesiveness of Obtained Powders

The bulk density (*ρ_b_*) and tapped density (*ρ_t_*) of the powders were determined by the ratio of their mass to the occupied volume, measured using a graduated cylinder before and after tapping the cylinder, respectively. The Carr index (*CaI*) and Hausner ratio (*HR*) were computed to assess the flowability and cohesiveness of the microspheres, as per Equations (1) and (2), respectively.
(1)$$CaI\left(\%\right)=\frac{\left(\rho_{t}-\rho_{b}\right)}{\rho_{t}}\times 100$$
(2)$$HR=\frac{\rho_{t}}{\rho_{b}}$$

### 2.9. Solubility, Hygroscopicity, and Wettability

For the measurement of water solubility, 1 g of powder was dispersed in 15 mL of distilled water, vortexed for 3 min, and then centrifuged at 4000 rpm for 15 min. The supernatant was collected and dried at 110 ± 5 °C until a constant weight was achieved. The percentage ratio of the dry weight of solubilized materials to the initial sample weight was considered as the solubility [25]. To measure hygroscopicity, a powder sample (predetermined weight) was placed in a desiccator (75% relative humidity obtained by saturated NaCl solution) for 7 days. The hygroscopicity was defined as weight gain (g) of water per 100 g sample. To assess wettability, an amount of 50 mg powder was sprinkled onto the water surface, and the time taken (in seconds) for complete submergence of the particles was recorded as the wettability parameter.

### 2.10. Field Emission Scanning Electron Microscopy (FE-SEM)

The surface morphology of spray-dried powders was characterized using SEM (MIRA3 TESCAN, Brno, Czech Republic). The powder samples were mounted on aluminum plates and coated with a thin layer of gold using a vacuum ion sputter (SC7620 sputter-coaters, Quorum Technologies, Laughton, UK) for SEM observation.

### 2.11. Color Analysis

The color characteristics of the spray-dried powders were evaluated using a digital image processing technique as outlined in a previously described method [26]. Color parameters were calculated using Equations (3)–(5):(3)$$C^{*}=\sqrt{{\left(a^{*}\right)}^{2}+{\left(b^{*}\right)}^{2}}$$
(4)$$WI=100-\sqrt{{\left(100-L^{*}\right)}^{2}+{\left(a^{*}\right)}^{2}+{\left(b^{*}\right)}^{2}}$$
(5)$$YI=\frac{142.86\times b^{*}}{L^{*}}$$
where *L** = lightness, *a** = redness/greenness, *b** = yellowness/blueness, and *C**, *WI*, and *YI* are chroma, whiteness, and yellowness indices of the samples, respectively.

### 2.12. Fourier Transform-Infrared Spectroscopy (FT-IR)

The powder samples were blended with potassium bromide (KBr) at a ratio of 1:100 and compressed into disk shapes. These sample discs were analyzed using an FT-IR spectrophotometer (Shimadzu 8400, Tokyo, Japan) in transmission mode across the wavenumber range of 400–4000 cm^−1^.

### 2.13. Thermal Properties

The thermal properties of free EO and spray-dried powders encapsulating EO were analyzed by a simultaneous thermal analyzer (SDT-Q600, TA instruments, Lindon, UT, USA). An amount of approximately 15 mg samples were transferred to aluminum pans, hermetically sealed and heated from 30 to 600 °C at 10 °C/min under a stream of Argon atmosphere with 50 mL/min flow rate.

### 2.14. Antioxidant Properties

The antioxidant capacity of spray-dried encapsulated microparticles, along with that of free EO, was measured using DPPH and ABTS^+^ radicals’ scavenging activity assays. A method by Shahi et al. [27] was used to measure the inhibition percent for both DPPH and ABTS^+^ radicals (Equation (6)), assuming that the absorptions for DPPH and ABTS^+^ radicals were read at 517 nm and 734 nm, respectively:(6)$$Inhibition\left(\%\right)=1-\frac{A_{sample}}{A_{blank}}\times 100$$

### 2.15. In-Vitro Release Behavior

The amount of EO released from spray-dried particles was assessed using dialysis bags in a buffer phosphate medium [28]. In this measurement, 40 mg of spray-dried powder was suspended in 4 mL of phosphate buffer at a pH of 7.3 (as release media), and the suspension was transferred to a dialysis bag (D0530, Sigma Aldrich, Hamburg, Germany) previously conditioned in the same media. The bag was immersed in the release media (20 mL) and kept at a constant temperature of 35 °C with gentle shaking for a while. To measure released EO, specific volumes of release media were taken (substituted with an equivalent volume of fresh buffer) at defined time intervals of up to 300 min. Then, UV–vis spectrometry (Varian Cary 500, Agilent Technologies, Santa Clara, CA, USA) at 270 nm was used to measure the amount of released EO from dried particles. The cumulative concentration of released EO was calculated during the whole release time.

### 2.16. Statistical Analyses

The impact of applying partial enzymatic hydrolysis or ultrasound treatment of WPC and using their binary combination with MD as wall material treatments on different properties of spray-dried microparticles was investigated through factorial experiments using a completely randomized design (CRD). Mean comparisons were conducted using Duncan’s test at a significance level of 5%. Statistical analyses were performed using SAS software version 9.2. All preparations and measurements were independently conducted in triplicate to ensure reliability.

## 3. Results and Discussion

### 3.1. Characterization of Essential Oil from Chavir

The yield of extracted EO from Chavir was 1.2 ± 0.1% (*v*/*w*, wet basis), similar to the values obtained in previous reports [4]. GC-MS analysis of EO identified 21 chemical compounds comprising 89.6% of the total chemical composition. Among other constituents of EO, cis-β-ocimene with 23.25%, α-pinene with 14.51%, bornyl acetate with 11.25%, and cis- and trans-verbenol with 7.17% were the most abundant compounds in the EO (Table 2). This finding aligns with the outcomes documented in previous studies [3,21,29].

### 3.2. Physical Characteristics of Emulsions Containing EO

The droplet size of emulsions containing *Ferulago angulata* EO prepared using MD and WPC (modified or unmodified) varied within a range of 203.2–1185.0 nm (Table 3). The enzyme hydrolysis of whey protein resulted in emulsions with slightly smaller droplet sizes (243–289 nm) than those prepared using untreated WPC with an emulsion droplet size of 253–338 nm. On the other hand, power ultrasonication of WPC led to an increase in droplet size of emulsions with higher amounts of EO, which was opposite to samples with lower EO content. Based on the results, it was evident that the droplet size of emulsions containing higher EO content was larger (289 nm in HWPC2 sample to 1185 nm in MD-HWPC2 sample, Table 3) compared with their corresponding samples with lower EO concentration, which fairly varied in the range of 203 nm in UWPC3 sample to 253 nm in MD-WPC3 sample. The largest size was achieved for binary systems of MD-HWPC2 (with 1185 nm) and MD-WPC2 (with 849 nm). Generally, binary emulsion samples were of larger droplet size in comparison with simple emulsion systems, more profoundly for the EO:WM ratio of 1:2. This could be an indication of effective interactions between MD and WPC, which has also been reported by others [30].

The consistency (uniformity) of droplet size in emulsions can be deduced from the polydispersity index (PDI), as shown in Table 3. The PDI results unveiled that almost all emulsions were of acceptable uniformity, the best observed for samples at an EO:WM ratio of 1:3, which were of smaller droplet size. Furthermore, modification of whey protein using both methods resulted in more uniform emulsions than those prepared by intact whey protein. The emulsions prepared by MD alone had the highest PDI values (0.512–0.630). Due to the lower PDI and droplet size of samples at emulsions with an EO:WM ratio of 1:3 it was decided to produce spray-dried particles with this EO:WM ratio.

Based on creaming index results presented in Table 3, it was found that all emulsion samples were stable without any measurable phase separation, except those samples prepared by MD, which had phase separation above 60%. The high instability of the MD-prepared emulsion could probably be related to the lack of surface-active properties of MD [31]. These results are consistent with the droplet size and polydispersity index (PDI) data obtained in this investigation. The stability of emulsions could be attributed to the enhanced emulsifying and emulsion stabilizing properties of HWPC and UWPC [17].

The charge density on the surface of emulsions characterized by Zeta potential (Table 3) revealed that all samples were of moderately high charge, inferring the stability of droplets in the continuous phase, as observed by the results of CI. Zeta potential values of UWPC emulsions were higher than those for intact WPC, which in turn were higher than HWPC emulsions. The Zeta potential values of modified whey proteins found in the present work were higher than those reported by Sarabandi et al. [13]. Generally, it can be postulated from Zeta potential data that blending MD with WPC reduces the charge density of particles, an indication of successful interactions between the two components. The larger the emulsion droplets, the lower the Zeta potential value.

The lowest encapsulation efficiency (EE) of essential oil was observed for emulsions prepared by MD (the most unstable sample), whereas the EE of binary emulsions was generally higher than 90% (Table 3), showing the most successful systems for encapsulation of EO. The emulsions with the best EE were MD-HWPC at both EO:WM ratios used. Although simple emulsions of WPC (intact and modified) had acceptable EE values, the EE increased at higher EO concentrations. The high EE values of binary blend systems have been attributed to the combined emulsification properties of MD and WPC [32], especially for modified WPCs.

### 3.3. Physical Characterization of Spray-Dried Microparticles

According to the results in Table 4, the largest particle sizes corresponded to binary blends of untreated WPC with MD (30.21 µm). Untreated sole MD and WPC as wall materials gave particles with an average size of 19.52–24.61 µm. Application of hydrolyzed (HWPC) and sonicated WPC (UWPC) led to the production of microparticles with the smallest size with an average diameter of 6.23 and 12.95 µm, respectively. However, the application of binary blends of MD with HWPC or UWPC increased their mean particle size to 8.24 µm and 13.5 µm, respectively. It was concluded from dynamic particle size measurement that enzymatic hydrolysis of WPC led to the formation of the smallest particles.

Moreover, the application of binary blends of MD with each of the treated WPC (both HWPC and UWPC) had an improving effect on particle size. Different particle sizes of samples can be attributed to the viscosity of the produced feed and, thereby, an increase in the size of the atomized droplets inside the chamber [33]. Spray-drying yield varied in a range of 64.1–85.0% for different powder samples. Higher yields were obtained for binary blends of MD with UWPC samples. Apparently, large particles, especially those produced from binary blends of carriers, resulted in more recovery in the drying chamber. Contrary to powders obtained from ultrasound-treated WPC, enzymatic hydrolysis of WPC led to the formation of small particles, resulting in less recovery from the drying chamber (yield = 64.1–67.0%).

The particle size distribution of spray-dried particles varied in the range of 0.31–0.65, which were of acceptable PDI values. Of course, there was a particular dependency of PDI values on the type of carriers used. Application of sonicated WPC (alone or in combination with MD) led to the production of microparticles with the least PDIs (0.31–0.33). This could be an indication of effective interactions between MD and modified WPCs as binary blends of wall materials, which has also been confirmed in other studies [30]. The results of Zeta potential (ZP) obtained for different spray-dried microparticles revealed that HWPC (alone or combined with MD) produced particles with the least surface charge (−18.5 to −20.3 mV). In contrast, sonication of WPC led to a higher surface charge with ZP values of −26.3 mV.

Interestingly, binary blends of UWPC with MD resulted in even more surface charge density with ZP values of −29.1 mV. This may be explained as an indication of successful interactions between sonicated WPC and MD in the atomization process in spray-drying. It has been reported that the difference in Zeta potential values might be attributed to the presence of depolymerized protein fractions with different solubility, composition of charged amino acids, surface activity, flexibility, and migration rate at the emulsion interface [34].

The lowest encapsulation efficiency of essential oil was observed for powders produced by hydrolyzed WPC alone or combined with MD (EE = 58.2–61.3%). The EE of binary systems obtained from UWPC (89.0%) was highest among SD samples, indicating the most successful covering systems for encapsulation of EO. The higher EE of binary blend systems has been attributed to the combined emulsification properties of MD and whey protein [32], especially with modified whey proteins [13].

Moisture is one of the effective indicators of the physical, oxidative, and microbial stability and maintaining the flow behavior of powders. The moisture value (2.86–3.42%) and water activity (0.31–0.36) of the samples indicated the appropriate physical, microbial and oxidative stability of spray-dried powders. In this regard, the solubility, molecular weight, degree of hydrophilicity/hydrophobicity, and water-holding capacity of treated (hydrolyzed or sonicated) proteins are factors affecting the amount of moisture and water activity of powders [35,36]. Solubility and hygroscopicity are other essential indicators of powders, especially in applications, and they improve the miscibility and compatibility of lipophilic compounds in the food system [37]. The highest amount of solubility (86.4%) and hygroscopicity (12.4%) were related to the MD-HWPC sample (Table 4). The lowest amounts of these indices corresponded to SD powders produced from MD or WPC alone. The values of these indices are significantly affected by the nature, structural characteristics, and the amount of hydrophilic areas available to the carrier (proteins) [33]. Other morphological, structural, porosity, and particle size indicators are also influential on the solubility of powders [38]. Hygroscopicity is usually affected by the molecular weight of the peptide/protein, the amount of surface oil, the hydrophilic and moisture absorbing areas on the surface of the particles, the chemical and structural nature of the carriers and, finally, the amount of moisture and the size of the particles [38]. The solubility and hygroscopicity of spray-dried microparticles from the MD-HWPC carrier were greater than those of other carriers. This phenomenon may be attributed to structural modifications in the HWPC, resulting in increased accessibility and exposure of hydrophilic regions.

Density (bulk/tapped) and angle of repose were other indices that may affect the packaging and storage of powders. In general, there is an inverse relationship between the angle of repose and the bulk/tapped densities—a lower angle of repose means higher densities due to better particle packing. A lower angle of repose indicates the particles flow more freely over each other, allowing for better packing and fewer air spaces when piled. Conversely, a powder with a higher angle of repose (worse flow) will have lower bulk/tapped densities. The particles cannot pack as tightly due to greater friction and interparticle forces. As shown in Table 4, the highest values of these indices were related to the samples produced with modified WPC (HWPC and UWPC) with values of 35.0°–36.7°, respectively. Carriers with modified WPCs decreased bulk/tapped densities of the spray powders (0.50–0.51 g/mL). This variation can be ascribed to discrepancies in structural features, surface morphology, particle size distribution, adhesion, and agglomeration, which ultimately result in diverse porosities. However, the decrease in bulk/tapped densities leads to an increase in the amount of air trapped between the particles and may result in the risk of lipid oxidation. However, the ability of the carriers to create an insulating film/coating within the matrix is a key factor in the oxidative stability of the particles [37,39]. Based on flowability data for spray-dried powders calculated using the Carr index and Hausner Ratio (Table 4), the flowability of all powders was satisfactory and remained unaffected by both the carrier type and any modifications made.

### 3.4. Morphological Properties Using FE-SEM Observations

Morphological evaluation of powders with the FE-SEM technique is one of the crucial observation methods to determine the physical stability of microparticles (occurrence of mechanisms such as Ostwald ripening, agglomeration, and integration) during storage [40]. The surface properties of microparticles during spray-drying are impacted by factors such as the composition and concentration of components in the wall material, the rate of moisture evaporation, the transfer of solids to the wall layer, and the formation of a crust [41]. Figure 1A–C shows the appearance and morphological characteristics of spray-dried microparticles stabilized with different wall materials: (A) intact WPC or combination of MD-WPC, (B) UWPC or combination of MD-UWPC, and (C) HWPC or combination of MD-HWPC.

The SEM study showed that the morphology of microparticles was influenced by the type and physical modification of the wall material. The powders produced with intact WPC were somehow irregular and wrinkled, showing the presence of indentations and some pores on the surface of particles (Figure 1A). At the same time, the powders obtained from a binary blend of MD and WPC had considerably improved surface morphology compared to that of intact WPC with more uniform shapes and sizes (with a diameter of 10–50 µm). These particles were relatively spherical with small surface grooves, sometimes showing tiny reservoir-type pores. Similar reports show that emulsion particles incorporating chia oil [42] and grape seed oil [13] stabilized by modified WPC exhibited spherical particles with uniform and regular surfaces. Applying UWPC (Figure 1B) as wall material in spray-drying produced microparticles with increased small cracks and holes on their surface. However, a combination of UWPC with MD as a feed solution led to the production of particles with a higher degree of homogeneity, relatively spherical structures with smooth surfaces, and no evidence of aggregation. This indicates that the formation of microcapsules was successful. Of all the wall material treatments tested in this study, the MD-UWPC system showed the most significant enhancement in the morphology of microparticles, exhibiting a consistent shape and size. This suggests that the essential oil as the core material was effectively encapsulated by the wall material system through the spray-drying process. Figure 1C shows that enzymatic hydrolysis of WPC negatively affected the integrity of microparticles. Most of the produced particles lost their spherical shapes. A higher magnification of the SEM images in Figure 1C(ii,iii) revealed the presence of broken microparticles with irregular shapes and surface cracks. These findings suggest that the process of microcapsule formation was inferior. Interestingly, the application of binary blends of MD with HWPC did not help in the recovery of spherical-shaped particles in the spray-drying process, and the corresponding SEM images clearly showed no improvement in the surface morphology of microparticles (Figure 1C). In other similar studies, the use of protein and polysaccharide carriers (alone or in combination) as emulsifiers and carriers in the microencapsulation of drumstick (*Moringa oleifera*) oil [38], European eel (*Anguilla anguilla*) oil [35], walnut oil [43], baltic herring (*Clupea harengus membras*) oil [44], fish oil [45], squalene [32], and chia oil [42] resulted in the generation of particles characterized by irregular, spherical structures with either smooth or indented surfaces, and walls that were either intact or broken, exhibiting numerous pores.

### 3.5. Color Analysis of Spray-Dried Powders

Color parameters determine the quality characteristics, conditions of use, and usability of a formula/or combination in food formulations. In this research, the effect of carrier modification and blends of MD with untreated or treated WPC on the color parameters of microparticles of encapsulated Chavir EO. Figure 2 demonstrates the color properties of microencapsulated EO powders obtained from different wall materials. Total color difference (ΔE) reflects the color differences of powders as compared to standard color plates (in this study, RAL 080 85 05 with the color name of white wheat flour). Treatments incorporating UWPC showed the highest ΔE values (19.1). In contrast, HWPC or MD-WPC corresponded to the lowest ΔEs (3.9–4.2), reflecting the lowest color difference concerning the standard RAL plate used in this study. Ultrasound-treated WPC created yellowish powders with yellowness indices (YI) of 40.2 compared to 17.3 for intact WPC. HWPC as wall material produced powders with a creamy to white appearance and a YI of 10.5. However, blending MD to those carriers before spray-drying significantly (*p* < 0.05) decreased YIs from 40.2 to 25.8, 17.3 to 11.2, and 10.45 to 5.0 for UWPC, WPC, and HWPC treatments, respectively, and the resulting powders appeared as creamy to white. Whiteness index (WI) data confirmed results as mentioned above in a way that WI values were increased with blends of MD to each of the other carriers. Also, the highest WI values corresponded to MD-HWPC and MD-WPC or MD-HWPC, with values of 91.6 and 82.9.

Chroma data can reflect the color of spray-dried powders by providing information about the intensity or purity of the color. Chroma, which represents the saturation or vividness of a color, can help in determining how vibrant or dull the color of the spray-dried powder is. By analyzing the chroma data of the powder samples, researchers can quantify the color attributes and assess factors such as color uniformity, stability, and quality. This information can be valuable in industries such as food, pharmaceuticals, and cosmetics, where color consistency is crucial for product quality and consumer acceptance. Chroma analysis showed the same trend observed and reported for ΔE values YI data. Microencapsulated particles based on HWPC or MD-HWPC showed the highest chroma values, and treatments with HWPC and MD-HWPC demonstrated the lowest values. Studies have indicated that the carrier type (whether chemically or physically modified) and other chemical processes, like pigment degradation and fatty acid oxidation, including browning reactions during spray-drying, may result in alterations to the color characteristics of the final powders [46].

### 3.6. Analysis of Chemical Structure of Spray-Dried Microparticles by FTIR

FTIR analysis was employed to assess the chemical structure and interactions among the components of neat FA essential oil (EO) and various wall materials. The results offer valuable insights into the interaction of EO within the structure of loaded microparticles. In this study, the chemical structures of wall material components (in forms of intact and modified) as well as core material (FA essential oil) in preparation of encapsulated microparticles were investigated. Figure 3 demonstrates FTIR peaks of neat FA essential oil (Figure 3A) and different spray-dried powders (Figure 3B). In general, the main structural regions of whey proteins were identified as follows [47]: (1) amide-A region (N-H stretch) at frequencies 3300–3420 cm^−1^ for WPC; (2) amide-B region (C-H and O-H stretch) at 2926 cm^−1^ frequency (for all intact or modified proteins); (3) amide-I region (C=O stretch) at 1650 cm^−1^; (4) amide-II (N-H deformation and C-N stretch at 1550 cm^−1^; (5) amide-III (including secondary structures of α-helix, β-sheet, β-turn, and random coils) at 1243 cm^−1^; and (6) other polysaccharide factors and C=O stretch in the frequency of 1030–1080 cm^−1^ and N-H bending at 600–650 cm^−1^.

FTIR spectrum of FA essential oil (Figure 3B) showed a broad absorption band observed at 3400–3500 cm^−1^ relating to C—H stretching of cis-alkene groups, stretching vibrations of cis double bands, —CH_2_ stretching, and bending [48,49]. There are also prominent peaks at 2925 cm^−1^ (associated with O—H, N—H, and C—H stretching of aldehydes), 1732 cm^−1^ (indicative of C = O and C = C stretching of aldehyde/ketones), and 1242 cm^−1^ (attributed to —CH_3_ and C—H in-plane bending). These peaks are characteristic of α-Pinene and β-Ocimene identified as the major constituents of FA essential oil, a finding supported by GC-MS analysis in this study (Table 2) and the previous literature [50,51]. Additionally, peaks at 1450 and 1242 cm^−1^ are linked to the methyl band and C—H vibration absorption of benzene rings, respectively [52].

Comparing FTIR spectra of neat EO (Figure 3A) with those of microencapsulated EO (Figure 3B) revealed the appearance of high-intensity peaks at 3300–3420 cm^−1^ (C—H stretching), suggesting the incorporation of EO oil within the carriers’ structure. Previous findings have suggested that this phenomenon could be linked to the effective encapsulation of essential oil within the carriers’ structure [53]. After the spray drying of essential oil with different carriers used in this study, several important changes occurred: (1) a sharp decrease in the intensity of EO peak in the regions of 2925 cm^−1^ after encapsulation as a result of potential formation of hydrogen bonds between the cis-alkene groups of fatty acids and the hydroxyl groups of the amide-A and B regions of proteins, along with the shielding of these regions within the protein matrix [35]; (2) a decrease in the intensity of the peak related to carbonyl ester groups (1732 cm^−1^) was observed after encapsulation (especially in the samples produced with WPC or MD-WPC). These changes are the result of trapping and placing of oil in the carrier matrix without chemical reaction between them [49]; (3) a significant increase in the intensity of absorption peaks at 1030 cm^−1^ after spray drying with almost all combinations of wall materials relating to the involvement of amide regions (I, II, and III) of the WPC proteins, which indicates the complete distribution and placement of the functional groups.

### 3.7. Thermal Characteristics of Spray-Dried Powders

Thermogravimetric analysis (TGA) provides essential information about the thermal stability, decomposition temperature, and residual mass of spray-dried samples by measuring the weight change as a function of temperature. Differential scanning calorimetry (DSC) is another powerful tool that offers insights into the thermal transitional events occurring in the sample by measuring the heat flow associated with those transitions. In this study, simultaneous thermal analysis (STA) provided concurrent measurement of thermal events using both TGA and DSC techniques. Furthermore, derivative thermogravimetry (DTG) in STA experiments offers additional information by analyzing the rate of weight change as a function of applied temperature, offering a more detailed insight into the thermal decomposition processes. By integrating TGA, DSC, and DTG in STA experiments, it is possible to gain a thorough understanding of the thermal properties of the microcapsules and identify key thermal parameters critical for their characterization.

Figure 4 demonstrates STA thermal diagrams of neat FA EO (Figure 4A), spray-dried microcapsules produced using MD-WPC (Figure 4B), MD-UWPC (Figure 4C), and MD-HWPC (Figure 4D). The temperature of maximum degradation rate (*T*_d_, °C), weight loss (ΔW, %), and weight residue of experimental samples are also shown in Table 5. In Figure 4A, a distinct and abrupt weight loss (WL) was noted, attributed to the thermal decomposition of pure essential oil within the temperature range of 55–110 °C, with a peak degradation temperature (Td) of 55 °C. This thermal process, resulting in a 95% reduction in the initial weight of the essential oil, was marked by an exothermic peak at around 155 °C, indicating a thermal decomposition event in the DSC thermogram of the essential oil. As reported in other studies, EO from different plants shows shallow thermal stability in TGA analysis [54,55]. In a general outcome from Figure 4, it can be seen that encapsulating EO using MD-WPC wall materials, regardless of carrier type, increased the thermal stability of EO from 55 °C to 180–250 °C. This indicates an enhancement of 2 to 2.5-fold in the degradation temperatures (*T*_d_) of EO within spray-dried microcapsules.

There is a three-step thermal degradation event in all spray-dried powders (Figure 4B–D). In the MD-WPC sample (Figure 4B), the first weight loss was minimal (2.1% of initial weight) observed at *T* below 100 °C. This phenomenon can be associated with the evaporation of moisture and residual essential oils adhering to the microparticles. This occurrence was supported by an initial endothermic peak (at 120–150 °C) observed in the corresponding DSC thermogram. The second weight loss was at *T*_d_ of 250 °C, which led to a 10.5% weight reduction due to thermal decomposition of microparticles. The third thermal event, occurring between 320 °C and 500 °C, resulted in a final weight loss of 7.5%, likely attributed to the thermal decomposition of the encapsulated essential oil within the MD-WPC carrier. This stage did not show any additional exothermic peak in the DSC analysis. As shown in Figure 4B, the MD-WPC sample is more stable in terms of thermal degradation than other samples. This can be proven by the remaining weight of almost 80% after STA analysis. Meanwhile, for MD-UWPC and MD-HWPC treatments, more weight losses were observed (ca. 55%), with a remaining total material of ca. 45% after thermal analysis. This difference can be related to the modification of WPC by either ultrasonication or enzymatic hydrolysis, which in turn may have led to the susceptibility of the carriers to thermal degradation.

STA thermal characterization of the MD-UWPC sample showed similar three-stage degradation profiles but at different weight loss and thermal degradation temperatures at three analysis steps, as shown in Table 5. However, the MD-HWPC sample demonstrated weaker thermal stability with a two-stage weight loss, demonstrating the loss of essential oil at earlier stages (*T*_d_ of 182 °C) with concomitant endothermic DSC event at this stage, indicating destruction of core material due to evaporation. This thermal behavior was not seen for MD-WPC or MD-UWPC samples, indicating that the MD-HWPC sample is susceptible to thermal decomposition. As shown in SEM images (Figure 1C), the latter sample already exhibited more open-structured microparticles, which may explain the lower thermal stability of these microparticles.

In summary, the thermal analysis results indicate that encapsulating unstable substances like essential oils into thermally stable microparticles produced through spray-drying using carriers such as MD-WPC (in either intact or modified forms) could enhance their thermal stability when compared to pure materials. Therefore, HWPC or UWPC, in combination with MD as a wall compound, could cover free EO as heat-sensitive compounds and boost their thermal endurance. Our results are in accordance with those reported by other researchers [56,57].

### 3.8. Total Phenolic Content and Antioxidant Properties

It is shown in Figure 5 that the neat EO extracted from *Ferulago angulate* (in its free form) exhibited the highest amount of TPC and DPPH^−^/ABTS^+^ radicals scavenging activity (RSA).

This is due to the presence of phenolic and antioxidant compounds such as ocimene (23.3%), α-pinene (14.5%), bornyl acetate (11.3%), and verbenol (7.2%) that were found in GC-MS analysis of FA EO shown in (Table 2). Stabilization of EO within spray-dried microcapsules using WPC (either intact or modified) significantly (*p* < 0.05) reduced its TPC and antioxidant activity. As reported in earlier studies, it is anticipated that there may be a decrease in the antioxidant activity of microencapsulated essential oil due to the stabilization process during spray-drying, which reduces its availability during antioxidant and TPC determination tests [13,58]. However, the type of WPC modification (enzyme hydrolysis vs. ultrasonication) and the use of binary blends of WPC-MD affected the available phenolic compounds and antioxidant activity of the produced microparticles. Among different spray-dried powders, those produced with HWPC or MD-HWPC as wall material showed significantly (*p* < 0.05) higher TPC and DPPH^−^/ABTS^+^ RSA values than UWPC and WPC carriers. This enormous difference in TPC and antioxidant activity of microparticles produced from HWPC could be related to their instability during and after the spray-drying process. As shown in SEM results (in Figure 1C), enzymatic hydrolysis of WPC negatively affected the integrity of spray-dried microparticles with the occurrence of broken particles with irregular shape and surface cracks, suggesting that the process of microcapsule formation was inferior. This could lead to the release of EO to the surface of microparticles, which, in turn, could increase the TPC and antioxidant activity of these particles. Figure 5. showed that the application of intact WPC (alone or in combination with MD) as wall materials produced particles with the lowest antioxidant properties because the EO cannot migrate to the surface of particles. In a previous study, it was shown that the application of power ultrasound and partial enzymatic hydrolysis of WPC increased DPPH^−^/ABTS^+^ radicals scavenging activities of spray-dried microcapsules incorporating grape seed oil compared to the situation where intact WPC were used as wall material [13]. These authors also showed that the hydrolyzed WPC had an even more significant effect in increasing the antioxidant properties of obtained microcapsules compared to sonicated WPC. These findings support our results in Figure 5, which show the enhanced antioxidant properties of HWPC samples compared to intact WPC or UWPC carriers.

### 3.9. Release Study of EO from Spray-Dried Microparticles

The cumulative release of EO content from spray-dried microparticles produced with different wall materials (WPC, UWPC, and HWPC) in phosphate buffer (pH = 7.3) was measured during 300 min, and the results are shown in Figure 6. This figure illustrates a biphasic release pattern for EO, wherein the release rate was rapid during the initial 50 min, followed by a gradual slowdown for the remainder of the measurement period. This pattern has been documented in other studies involving chitosan nanoparticles and microspheres as well [6,20]. The initial phase in the release profile is likely due to the swelling mechanism of the microparticles in the phosphate buffer [59], leading to a rapid release rate. The subsequent phase of the release profile may be associated with surface erosion and/or diffusion mechanisms, representing a slower process that results in a reduced release of the encapsulated essential oil. In this study, the lowest release content was observed for microparticles prepared with WPC and UWPC as wall material. This low-release content is probably attributed to the formation of stable microparticles (with denser encapsulating layer) by those carriers, which could retain the EO from escaping into release media [60]. However, the release content of spray-dried powders prepared by HWPC was significantly (*p* < 0.05) higher than other wall material treatments. This distinct difference in WPC is related to instability, such as microparticles. As shown in SEM results (in Figure 1C), enzymatic hydrolysis of WPC negatively affected the integrity of spray-dried microparticles with the occurrence of broken particles with irregular shape and surface cracks, indicating that the process of microcapsule formation was inferior. This could lead to increased release of EO to the surface of these microparticles. Moreover, successful encapsulation occurs by using wall materials with strong film-forming properties. Since the modifications applied in the current study disrupted the molecular structure of whey protein, it could be expected that the film-forming ability has probably been lost, resulting in a disrupted molecular structure and, thus, less integrity of the capsule surface, as observed in SEM micrographs. Overall, the low integrity of the microparticles’ surface led to higher and faster release in an aqueous medium, as was evident in the release kinetics results.

## 4. Conclusions

This study focused on investigating the efficiency of binary blends of modified WPC and MD in spray-drying encapsulation of *Ferulago angulata* essential oil. The choice of wall materials and their modifications (via power ultrasound or enzymatic hydrolysis) significantly influenced the physical characteristics and encapsulation efficiency of emulsions containing essential oils. The physical characterization of spray-dried microparticles revealed essential insights into particle size, yield, distribution, surface charge, encapsulation efficiency, moisture content, solubility, hygroscopicity, and flowability, emphasizing the impact of carrier type and modification on the physical properties of microparticles. Binary blends of modified WPC-MD showed improvements in particle sizes and yields, particularly with UWPC. Zeta potential measurements highlighted differences in surface charge density, with UWPC-MD blends exhibiting superior values. The SEM images revealed distinct differences in the surface characteristics of the particles based on the type and modification of the wall materials used. Thermal analysis elucidated the thermal stability of microcapsules, emphasizing the role of wall materials in enhancing thermal endurance. Evaluation of total phenolic content and antioxidant properties showcased the influence of carrier selection on the availability of phenolic compounds and antioxidant activity. The release study unveiled a biphasic release pattern of essential oil, with wall materials playing a crucial role in release kinetics. Microparticles prepared with HWPC exhibited higher release content, attributed to their instability, while enzymatic hydrolysis of WPC negatively impacted microparticle integrity, potentially increasing EO release. These findings collectively emphasize the pivotal role of wall materials in shaping the physical, morphological, thermal, antioxidant, and release properties of spray-dried microparticles, offering valuable insights for the development of microencapsulation techniques for essential oils.

## Figures and Tables

**Figure 1 foods-13-01407-f001:**
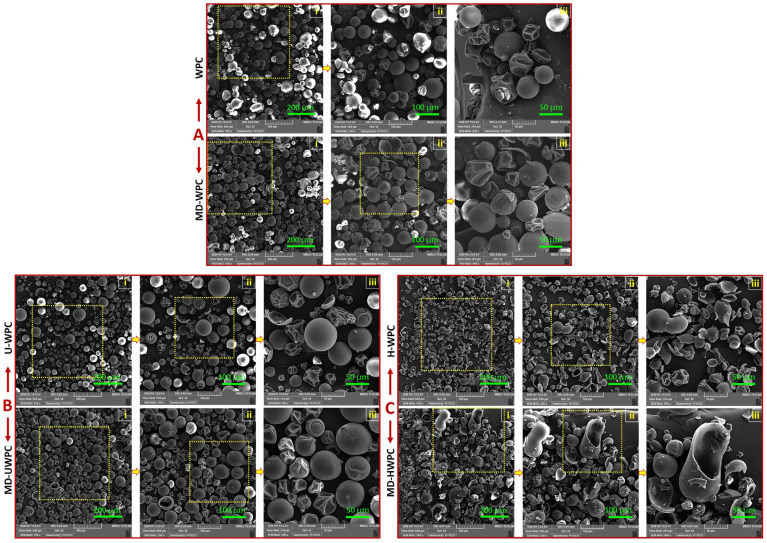
SEM images of spray-dried powders produced with (**A**) intact WPC and combination of MD-WPC, (**B**) ultrasound-treated WPC (UWPC) and combination of MD-UWPC, and (**C**) hydrolyzed WPC (HWPC) and combination of MD-HWPC. Images ‘ii’ and ‘iii’ are sequential magnifications of image ‘i’ in each row driven from yellow-dotted square areas.

**Figure 2 foods-13-01407-f002:**
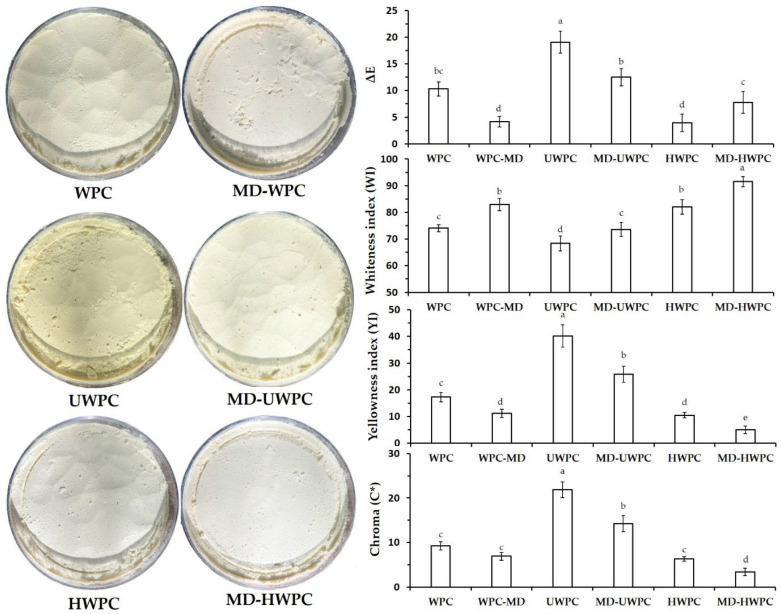
Digital images representing visual appearance (left-hand side) and different color parameters (right-hand side) of spray-dried powders produced using different carriers: WPC, MD-WPC, ultrasound-treated WPC (UWPC), MD-UWPC, hydrolyzed WPC (HWPC), and MD-HWPC blends. Data represent mean values of triplicate measurements. Error bars indicate SD values. Different alphabetical letters show significant differences between means (*p* < 0.05).

**Figure 3 foods-13-01407-f003:**
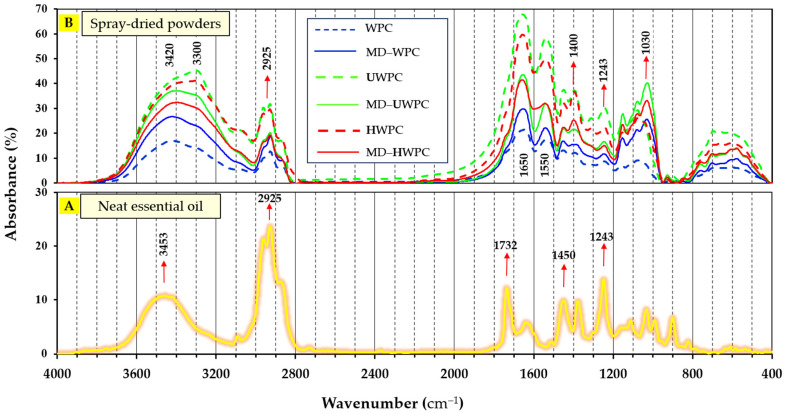
FTIR spectra of (**A**) neat *Ferulago angulata* essential oil, (**B**) spray-dried microcapsules produced by different carriers (WPC, MD-WPC, ultrasound-treated WPC (UWPC), MD-UWPC, hydrolyzed WPC (HWPC), and MD-HWPC blends.

**Figure 4 foods-13-01407-f004:**
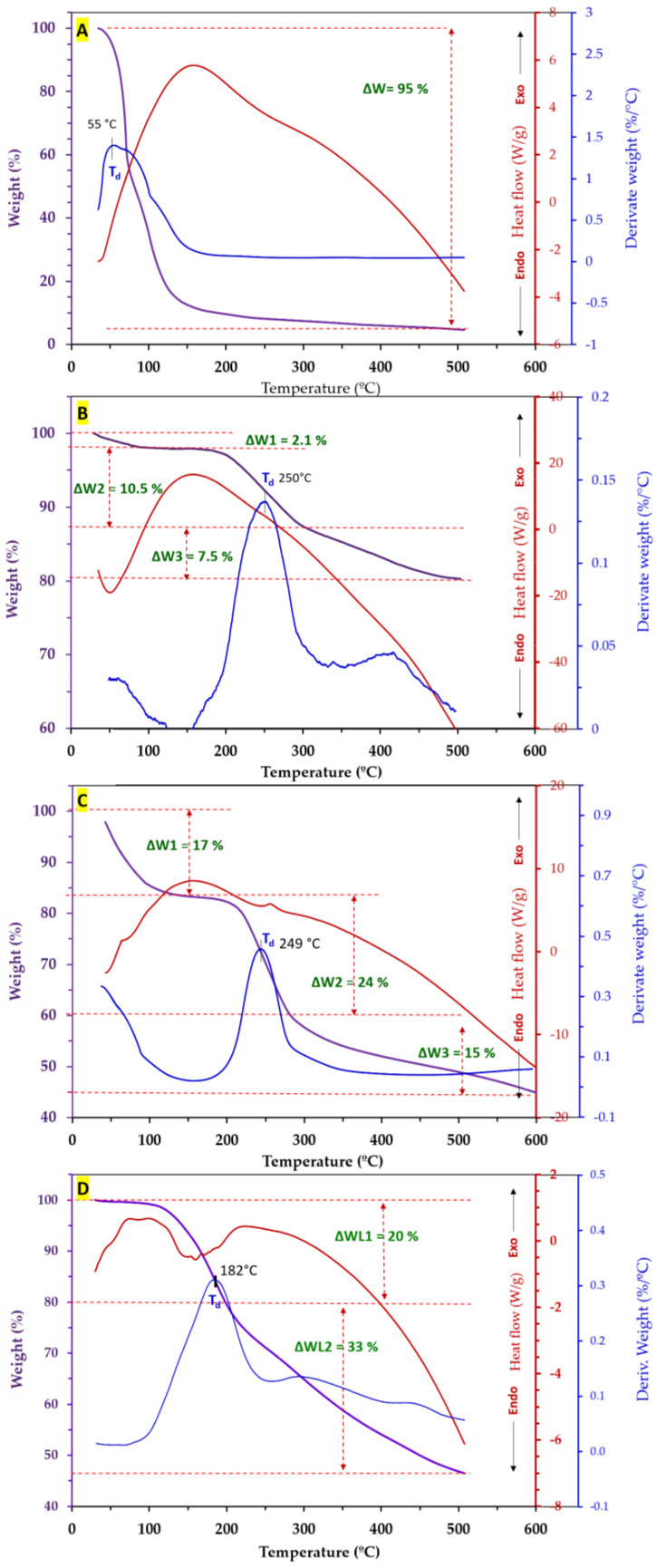
STA thermograms of neat *Ferulago angulate* EO (**A**) and encapsulated EO using different carriers: MD–WPC (**B**), MD–UWPC (**C**), and MD–HWPC (**D**).

**Figure 5 foods-13-01407-f005:**
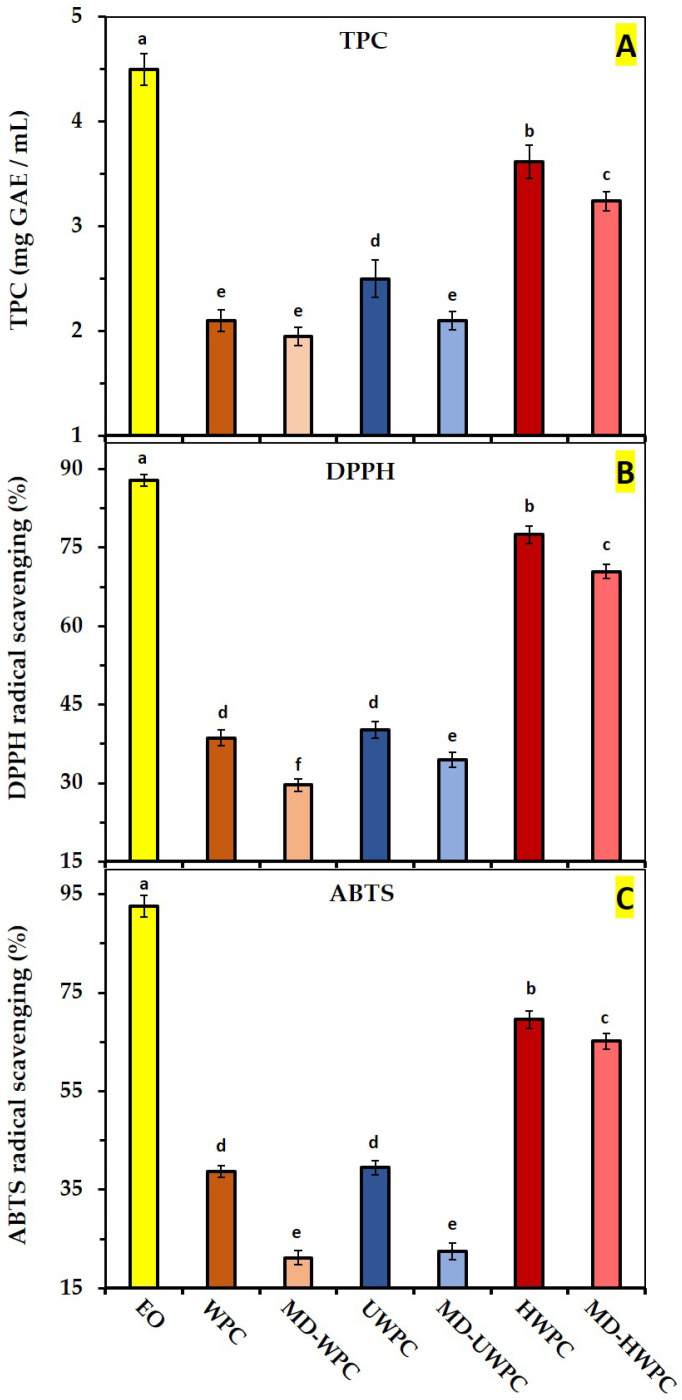
TPC (**A**), DPPH (**B**), and ABTS (**C**) radicals scavenging activities of neat *Ferulago angulate* EO (**A**) and its encapsulated form using different wall materials during the spray-drying process. Data represent mean values of triplicate measurements. Error bars indicate SD values. Different alphabetical letters show significant differences between means (*p* < 0.05).

**Figure 6 foods-13-01407-f006:**
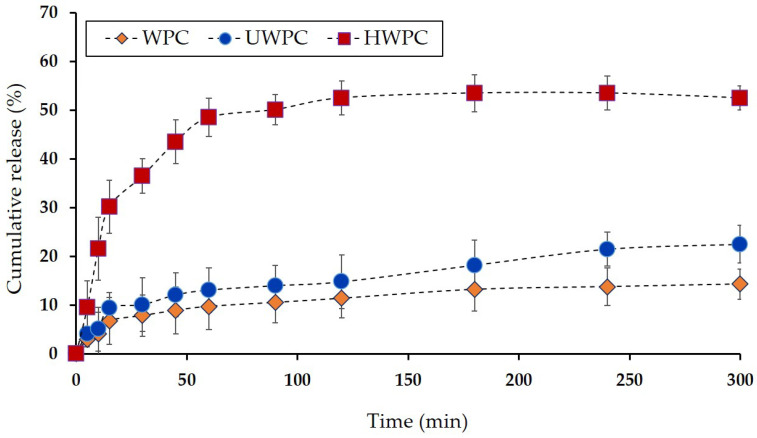
Release profile of *Ferulago angulate* EO from spray-dried microparticles produced with different wall materials (WPC, UWPC, and HWPC) in phosphate buffer (pH = 7.3). Data represent mean values of triplicate measurements, and error bars indicate SD values.

**Table 1 foods-13-01407-t001:** The formulations of emulsions prepared using different wall materials (WM) and EO ratios.

Sample Code	Wall Materials	EO:WM Ratio
MD2	Maltodextrin	1:2
MD3	Maltodextrin	1:3
WPC2	Whey protein concentrate	1:2
WPC3	Whey protein concentrate	1:3
HWPC2	Hydrolyzed WPC	1:2
HWPC3	Hydrolyzed WPC	1:3
UWPC2	Ultrasound-treated WPC	1:2
UWPC3	Ultrasound-treated WPC	1:3
MD-WPC2	Maltodextrin + WPC	1:2
MD-WPC3	Maltodextrin + WPC	1:3
MD-HWPC2	Maltodextrin + hydrolyzed WPC	1:2
MD-HWPC3	Maltodextrin + hydrolyzed WPC	1:3
MD-UWPC2	Maltodextrin + ultrasound-treated WPC	1:2
MD-UWPC3	Maltodextrin + ultrasound-treated WPC	1:3

**Table 2 foods-13-01407-t002:** Chemical composition of EO from Chavir (*Ferulago angulate*).

Component	Retention Time	Percentage
α-Pinene	3.95	14.51
Camphene	4.43	3.66
β-Pinene	5.02	3.27
β-Myrcene	5.41	3.96
δ-3-Carene	5.75	1.31
D-Limonene	6.38	3.41
cis-β-Ocimene	6.98	23.25
γ-Terpinene	7.30	4.17
Linalool	8.53	2.27
allo-Ocimene	8.97	2.13
cis-Verbenol	9.95	1.85
trans-Verbenol	10.29	5.32
Borneol	10.79	1.10
α-Terpineol	10.80	0.79
α-Phellandren-8-ol	11.10	1.19
Bornyl acetate	13.20	11.25
α-Muurolene	17.15	2.47
Methyl eugenol	17.47	0.89
Cadinol	18.10	0.45
β-Caryophyllene	22.35	0.58
Spathulenol	20.33	1.76
Total		89.59

**Table 3 foods-13-01407-t003:** Physical characteristics of emulsions with different concentrations of *Ferulago angulata* EO.

Sample Code	Droplet Size (nm)	PDI *	Zeta Potential (mV)	CI (%) **	EE (%) ***
MD2	294.2	0.512	−11.6	87	12.2
MD3	213.0	0.630	−20.9	64	15.8
WPC2	338.2	0.475	−30.6	nd	89.6
WPC3	252.7	0.282	−21.5	nd	60.5
HWPC2	289.4	0.368	−17.7	nd	90.1
HWPC3	242.7	0.345	−21.6	nd	78.7
UWPC2	583.4	0.572	−36.1	nd	90.0
UWPC3	203.2	0.238	−23.6	nd	76.3
MD-WPC2	848.8	0.601	−18.0	nd	91.6
MD-WPC3	253.4	0.365	−25.1	nd	91.3
MD-HWPC2	1185.0	0.374	−25.2	nd	93.5
MD-HWPC3	251.2	0.271	−27.4	nd	93.0
MD-UWPC2	655.4	0.518	−38.3	nd	92.4
MD-UWPC3	230.3	0.268	−29.2	nd	90.2

* PDI—polydispersity index; ** CI—creaming index; *** EE—encapsulation efficiency; nd—not determined.

**Table 4 foods-13-01407-t004:** Production yield and physical-functional properties of spray-dried powders containing EO.

Sample Code *	Yield (%)	Mean Particle Size (µm)	PDI	Zeta Potential (mV)	EE (%)
MD	75.1 ± 2.5 ^b^	19.52 ± 4.50 ^b^	0.60 ± 0.05 ^ab^	−25.7 ± 0.45 ^c^	78.9 ± 1.1 ^c^
WPC	79.0 ± 1.7 ^a^	24.61 ± 3.19 ^b^	0.65 ± 0.08 ^a^	−24.9 ± 0.13 ^c^	80.0 ± 1.5 ^c^
HWPC	64.1 ± 2.8 ^c^	6.23 ± 0.55 ^e^	0.55 ± 0.05 ^b^	−18.5 ± 0.12 ^e^	58.2 ± 1.6 ^e^
UWPC	79.2 ± 1.1 ^b^	10.95 ± 0.51 ^d^	0.33 ± 0.06 ^c^	−26.3 ± 0.18 ^b^	85.3 ± 1.9 ^b^
MD-WPC	82.3 ± 2.1 ^a^	30.21 ± 4.52 ^a^	0.65 ± 0.03 ^a^	−25.5 ± 0.12 ^c^	80.2 ± 0.8 ^c^
MD-HWPC	67.0 ± 1.2 ^c^	7.24 ± 0.85 ^e^	0.53 ± 0.05 ^b^	−20.3 ± 0.21 ^d^	61.3 ± 1.0 ^d^
MD-UWPC	85.0 ± 2.7 ^a^	12.92 ± 0.81 ^c^	0.31 ± 0.07 ^c^	−29.1 ± 0.37 ^a^	89.0 ± 1.2 ^a^
	Bulk density (g/mL)	Tapped density (g/mL)	Angle of repose (°)	Hausner ratio	Carr index
MD	0.55 ± 0.01 ^a^	0.65 ± 0.02 ^a^	31.3 ± 0.7 ^c^	1.15 ± 0.07 ^a^	0.13 ± 0.05 ^a^
WPC	0.48 ± 0.02 ^b^	0.58 ± 0.02 ^b^	33.3 ± 0.5 ^b^	1.19 ± 0.08 ^a^	0.14 ± 0.06 ^a^
HWPC	0.44 ± 0.01 ^c^	0.50 ± 0.01 ^c^	35.0 ± 0.4 ^a^	1.19 ± 0.05 ^a^	0.17 ± 0.07 ^a^
UWPC	0.42 ± 0.02 ^c^	0.51 ± 0.02 ^c^	36.7 ± 0.3 ^a^	1.21 ± 0.04 ^a^	0.17 ± 0.08 ^a^
MD-WPC	0.58 ± 0.02 ^a^	0.62 ± 0.01 ^a^	30.8 ± 1.4 ^c^	1.14 ± 0.02 ^a^	0.12 ± 0.04 ^a^
MD-HWPC	0.55 ± 0.03 ^a^	0.59 ± 0.02 ^a^	31.3 ± 0.5 ^c^	1.27 ± 0.10 ^a^	0.21 ± 0.05 ^a^
MD-UWPC	0.53 ± 0.02 ^a^	0.63 ± 0.02 ^a^	31.7 ± 0.7 ^c^	1.18 ± 0.03 ^a^	0.15 ± 0.07 ^a^
	Moisture content (%)	a_w_	Solubility (%)	Hygroscopicity (%)	
MD	3.11 ± 0.11 ^a^	0.33 ± 0.03 ^a^	74.90 ± 3.51 ^c^	9.15 ± 0.31 ^d^	
WPC	2.86 ± 0.05 ^a^	0.34 ± 0.01 ^a^	75.79 ± 2.19 ^c^	9.17 ± 0.40 ^d^	
HWPC	3.42 ± 0.05 ^a^	0.31 ± 0.02 ^a^	78.70 ± 1.92 ^c^	10.21 ± 0.24 ^c^	
UWPC	3.07 ± 0.08 ^a^	0.36 ± 0.02 ^a^	75.54 ± 2.23 ^c^	9.66 ± 0.42 ^d^	
MD-WPC	3.23 ± 0.06 ^a^	0.33 ± 0.01 ^a^	81.34 ± 1.58 ^b^	10.78 ± 0.72 ^b^	
MD-HWPC	3.04 ± 0.10 ^a^	0.35 ± 0.01 ^a^	86.35 ± 0.81 ^a^	12.36 ± 0.16 ^a^	
MD-UWPC	3.13 ± 0.05 ^a^	0.32 ± 0.01 ^a^	79.71 ± 0.82 ^b^	11.94 ± 0.36 ^b^	

* Sample codes are explained in Table 1. PDI—polydispersity index; EE—encapsulation efficiency. Data are represented as means of triplicate measurements ± SD. Different alphabetical letters show significant differences between means (*p* < 0.05).

**Table 5 foods-13-01407-t005:** Thermal and weight transition phases (Δ), degradation temperature (*T*_d_), and weight loss (ΔW) parameters of TGA-DTG-DSC thermograms for experimental materials subjected to thermal analysis.

Samples	Δ1		Δ2		Δ3		Weight Residue after Thermal Analysis (%)
	*T*_d__1, onset_ (°C)	ΔW_1_ (%)	*T*_d__2, onset_ (°C)	ΔW_2_ (%)	*T*_d__3, onset_ (°C)	ΔW_3_ (%)	
Neat EO	55–100	95.0	-	-	-	-	5.0
MD–WPC	55–100	2.1	250	10.5	400	7.5	81.0
MD–UWPC	55–100	17.0	249	24.0	400	15.0	44.0
MD–HWPC	55–100	20.0	182	33.0	-	-	47.0

## Data Availability

The original contributions presented in the study are included in the article, further inquiries can be directed to the corresponding author.

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
