# Peer review of "Enhancing Encapsulation Efficiency of Chavir Essential Oil via Enzymatic Hydrolysis and Ultrasonication of Whey Protein Concentrate–Maltodextrin"

_foods, 2024, doi:10.3390/foods13091407_

Round 1
Reviewer 1 Report
Comments and Suggestions for Authors
This paper includes detailed information related to the comprehensive study of the physicochemical and morphological characterization, thermal analysis, antioxidant properties, and release behavior of spray-dried microparticles encapsulating herbal essential oil using modified WPC and its binary combination with maltodextrin (MD) as wall materials. The subject is good and the novelty of the study is well defined.
Other comments:
1. Lines 50, Provide full name of HPP and PEF.
2. In Table 4, Fig.2, and Fig.5, What does the lowercase superscript stand for?
3. In antioxidant properties, was the amount of essential oil added consistent?
4. After the spray-dried microencapsulation, whether there was a change in the composition of essential oil content?
5. What were the chroma of WPC, MD, UWPC and HWPC wall materials?
Reviewer 2 Report
Comments and Suggestions for Authors
I find the work interesting and based on a wide range of empirical methods. Both sample preparation and the application of the mentioned methods are presented excellently. This is particularly important for the reader because it allows him to understand what a given method introduces and its limitations.
My critical – but not fundamental criticism – comments concern the Introduction and Conclusions sections. In the first one, the authors should relate the examined material in the context of other similar materials, which also applies to the lack of related references. This is important because the studied base material (I do not mean encapsulation) is interesting but relatively local and a broader reference to its properties, especially in the context of other similar materials that are more widely known. It is important for readers.
The second issue is related to interesting recalling applications of innovative non-thermal methods (HPP, PEF, ....). Today, they are primarily used for microbiological preservation and later for process property change. This interesting point requires a further – several sentences – comment and an explanation of its significance, or lack of relevance, for a given work.
Conclusions are too brief and do not reflect the 'richness' of the results nor provide a satisfactory discussion of the new conclusions arising from them. This section requires expanding, reconstructing, and showing the conclusions resulting from, but going beyond, the experimental results of the report. I mention this because these perspectives are visible in the results and are too sparingly used in the conclusions.
Reviewer 3 Report
Comments and Suggestions for Authors
#abstract Keep the abstract section brief and provide more information.
Line #50 need abbreviation.
Line #62 Maltodextrins (MD). please explain first.
Line #64 why combination needed ?
Line #70 aim is not clear. Please explain briefly.
Line #76 collection method not clear. How you take the sample ? How many person involve ?
Line #185 need reference
#Result section need to change. Every headline need one massage towards the reader.
#Figure 1. this is look like raw data. You need show manuscript way data for example crop and use scale bar. please change.
#Figure 2 & 5 & 6 need statistic analysis
#Conclusion need more information.
Comments on the Quality of English Language
ok
Reviewer 4 Report
Comments and Suggestions for Authors
Please see the comment in attached file.

Reviewer 5 Report
Comments and Suggestions for Authors
The study presents valuable research into the use of innovative carrier systems for the microencapsulation of essential oils, providing insights that could benefit the field significantly. However, there are several areas where clarification and further substantiation of claims would enhance the manuscript's impact and scientific rigor. Below, I have outlined key comments that should be addressed to improve the quality of the manuscript:
Introduction:
Lines 47-60: Could you clarify how the generalizations about the effects of various non-thermal techniques on proteins directly relate to their use in microencapsulation of essential oils? For instance, when stating that "Application of high power ultrasound would enhance the surface activity of most proteins leading to improved foaming and emulsifying properties" (Lines 51-52), can you provide specific examples or evidence from this study that demonstrate these effects in the context of essential oil microencapsulation?
Lines 68-69: In your assertion that "there is no comprehensive research on the application of power ultrasound- or enzymatically-modified WPC as wall-materials for spray drying of EO-loaded microspheres," could you specify whether this gap in research pertains to a particular aspect of EO microencapsulation or is it a broader claim? Would it be more accurate to state there is "limited research" rather than "no comprehensive research," unless it is clear that such studies are indeed absent in this scientific field?
Results and Discussion:
Lines 302-305: The statement regarding larger particles produced from binary blends resulting in more recovery in the drying chamber contrasts with the claim that smaller particles from enzymatically hydrolyzed WPC lead to less recovery. This seems counterintuitive unless explicitly supported by data on particle recovery dynamics, which is not detailed enough here.
Lines 351-352: The claim that structural modifications in HWPC result in increased accessibility and exposure of hydrophilic regions is broad and lacks specific empirical data linking these structural changes directly to the observed solubility and hygroscopicity outcomes. More detailed analysis or references are needed to substantiate this claim effectively.
Lines 567-572: The discussion on the thermal stability enhancement of essential oils in microparticles suggests a significant improvement without adequately discussing the chemical or physical interactions responsible for this stability. While it's stated that the encapsulation process contributes to this, the specific mechanisms or evidence from the study supporting this enhancement are not clearly elaborated.
Lines 598-607: The comparison of antioxidant properties between different treatments seems somewhat generalized. The statement that broken microparticles could lead to increased antioxidant activity due to EO release needs further clarification and empirical support. It's critical to differentiate whether the observed antioxidant activity is due to the release or the inherent properties of the wall materials used.
Lines 625-630: The assertion that enzymatic hydrolysis of WPC leads to instability and thus higher EO release is mentioned without discussing why this instability occurs at a molecular or structural level. This section lacks detailed analysis that would help in understanding the causative factors of increased release rates from these specific microparticles.
Comments on the Quality of English Language
Minor editing of English language required.
